# Diagnostic Implications of Irritable Bowel Syndrome Is an Independent Risk Factor for Undergoing Surgical Interventions in Patients with Inflammatory Bowel Disease

**DOI:** 10.3390/diagnostics13111901

**Published:** 2023-05-29

**Authors:** Yuhan Fu, Nisheet Waghray, Ronnie Fass, Gengqing Song

**Affiliations:** 1Department of Internal Medicine, MetroHealth Medical Center, Case Western Reserve University, Cleveland, OH 44109, USA; yfu@metrohealth.org; 2Department of Gastroenterology, MetroHealth Medical Center, Case Western Reserve University, Cleveland, OH 44109, USA; nwaghray@metrohealth.org (N.W.); rfass@metrohealth.org (R.F.)

**Keywords:** irritable bowel syndrome, ulcerative colitis, Crohn’s disease, surgeries

## Abstract

Background: Inflammatory bowel disease (IBD) and irritable bowel syndrome (IBS) can present with overlapping symptoms, making diagnosis and management challenging. Patients with IBD in remission may continue to experience IBS symptoms. Patients with IBS were found to have a disproportionately higher prevalence of abdominal and pelvic surgeries than the general population. Aims: The aim of this study was to determine whether IBS is a risk factor for undergoing surgical interventions in patients with IBD and explore the diagnostic implications of these findings. Methods: A population-based cohort analysis was performed using TriNetX. Patients with Crohn’s disease + IBS (CD + IBS) and ulcerative colitis + IBS (UC + IBS) were identified. The control groups consisted of patients with CD or UC alone without IBS. The main outcome was to compare the risks of undergoing surgical interventions between the cohorts. The secondary outcomes were to compare the risks of developing gastrointestinal symptoms and IBD-related complications between the cohorts. Results: Patients with IBD who subsequently developed IBS were more likely to experience gastrointestinal symptoms than those without IBS (*p* < 0.0001). Patients with concomitant IBD and IBS were more likely to develop IBD-related complications, including perforation of the intestine, gastrointestinal bleeding, colon cancer, and abdominal abscess (*p* < 0.05). Patients with concomitant IBD and IBS were more likely to undergo surgical interventions than patients without IBS, including colectomy, appendectomy, cholecystectomy, exploratory laparotomy, and hysterectomy (*p* < 0.05). Conclusions: IBS appears to be an independent risk factor for patients with IBD to develop IBD-related complications and undergo surgical interventions. Patients with concomitant IBD and IBS could represent a unique subgroup of IBD patients with more severe symptoms, highlighting the importance of accurate diagnosis and management in this population.

## 1. Introduction

Inflammatory bowel disease (IBD) is characterized by significant mucosal inflammation during the acute phase, whereas irritable bowel syndrome (IBS) is differentiated by normal mucosa and is considered to be a disorder of gut–brain interaction [1]. It has been increasingly recognized that IBD and IBS can be overlapping diseases, especially with postinfectious IBS [2,3]. Patients with IBD in remission can continue to exhibit IBS-like symptoms, such as abdominal pain, bloating, and altered bowel function. Accurate diagnosis can be challenging in these cases, as IBD patients in remission with concomitant IBS symptoms can be mislabeled as having active IBD, resulting in additional diagnostic testing. The overall prevalence of IBS in IBD patients was 32.5%; specifically, the prevalence of IBS in Crohn’s disease (CD) and ulcerative colitis (UC) in remission was estimated to be 36.6% and 28.7%, respectively [4]. Several possible mechanisms of overlapping IBD-IBS have been proposed, including disorders of the bidirectional interaction of the brain–gut axis [5,6], increased intestinal permeability [7,8], increased mucosal serotonin levels [9,10], psychological stresses [11,12,13], enteric nervous system dysfunction [14], and dysbiosis of gut microbiota [2,15]. Patients with concomitant IBD-IBS were also reported to suffer from a higher prevalence of psychiatric disorders, such as anxiety and depression [16]. Management of concomitant IBD and IBS can be very challenging, and there are very few clinical trials investigating possible treatment options for overlapping IBD-IBS. Most registered studies on clinical trials.org involved dietary and behavioral modifications rather than targeted pharmacological therapy [17,18].

Accurate diagnosis of concomitant IBD and IBS is crucial for guiding appropriate surgical and medical management. IBS is classified into four categories: IBS with predominant constipation (IBS-C), IBS with predominant diarrhea (IBS-D), IBS with mixed bowel habits, and IBS unclassified [19]. Its diagnosis is mainly based on symptoms fulfilling criteria and exclusion of organic etiology based on Rome IV criteria [20]. Diagnosis of IBD normally relies on endoscopic evidence of inflammation. However, in the real world, differentiating and approaching functional abdominal symptoms in IBD can be challenging. Accurate history-taking can be crucial in differentiating IBD and IBS. Serum inflammatory markers, inflammatory stool marker fecal calprotectin, and stool studies are commonly required [21]. Imaging and endoscopy are normally performed to assess the degree of inflammation [22]. Other possible etiologies of gastrointestinal symptoms should be ruled out when evaluating for functional symptoms in patients with IBD, such as small intestinal bacterial overgrowth, bile acid diarrhea, lactose intolerance, and chronic pancreatitis.

Surgical interventions are sometimes warranted for patients with IBD. Proctocolectomy and ileal pouch anastomosis are common surgical approaches for treating UC. Major surgical indications for UC are refractory UC, acute severe UC, severe UC-related complications, and UC-associated colorectal cancer [23]. Surgical management of CD is more challenging and depends on disease involvement, and is commonly reserved for patients with CD-related complications and refractory CD [24]. Small bowel resection is the most common surgical procedure for treating CD. On the other hand, patients with IBS were found to have a disproportionately high prevalence of abdominal and pelvic surgeries than the general population, such as cholecystectomy, appendectomy, and hysterectomy [25,26]. However, there are no prior studies that compared the risks of undergoing surgical interventions in IBD patients with concomitant IBS. Since IBS can complicate symptom presentation and clinical assessment, it is unclear whether surgical procedures will be performed more often or not in patients with concomitant IBD and IBS. 

The primary aim of this study was to determine whether IBS is an independent risk factor for undergoing surgical interventions in patients with IBD, with a focus on the diagnostic implications of these findings. The secondary outcomes were to compare the risks of developing gastrointestinal symptoms and IBD-related complications in patients with concomitant IBD and IBS.

## 2. Material and Methods

### 2.1. Data Source

TriNetX is a large global health research network containing de-identified aggregated electronic healthcare records (EHR) from 120 million patients around the world. There are 93 large healthcare organizations (HCOs) in the Global Collaborative Network, the majority of which are from the US and Europe. This platform provides comprehensive real-world healthcare information and allows for customized cohort selection. It contains diagnoses, procedures, medications, and laboratory values from both inpatient and outpatient settings. Review and approval by MetroHealth Medical Center Institutional Review Board (IRB) were exempted because there was no involvement of any identifiable patients’ personal information.

### 2.2. Cohort Definition

A population-based cohort study was performed by enrolling patients from the Global Collaborative Network in the TriNetX database. We identified 2 separate study populations: concomitant CD and IBS (CD + IBS) and concomitant UC and IBS (UC + IBS). There were 2 control groups: CD, and UC (Figure 1).

CD/UC + IBS cohorts: We identified all patients who had a colonoscopy (UMLS:SNOMED:73761001 or UMLS:CPT:1022231) within 1 year before the onset of CD (UMLS:ICD10CM:K50) or UC (UMLS:ICD10CM:K51). IBS (UMLS:ICD10CM:K58) diagnosis happened within 5 years after the onset of CD or UC. A normal stool calprotectin (UMLS:LNC:38445-3, between 0.00 and 200.00 µg/g), erythrocyte sedimentation rate (ESR, TNX:9066, between 0.00 and 20.00 mm/h) or C-reactive protein (CRP, TNX:9063, between 0.00 and 10.00 mg/L) level within 3 months before or after the diagnosis of IBS was required to ensure that IBD was in remission when IBS was diagnosed. 

CD/UC cohorts: Two control groups consist of patients with CD or UC alone without IBS. We identified all patients who had a colonoscopy within 1 year before the diagnosis of CD or UC. Patients with a diagnosis of IBS were excluded from the cohorts.

### 2.3. Outcome Measures

We collected baseline patient characteristics in each cohort, including average age at the onset of CD or UC, gender, race, and ethnicity. The primary outcome was to compare the 5-year risks of undergoing surgical interventions in CD/UC + IBS and CD/UC cohorts after propensity score matching was performed. Surgical interventions included small bowel resection, colectomy, appendectomy, cholecystectomy, exploratory laparotomy, and hysterectomy. We also compared the risks of developing various gastrointestinal symptoms within 5 years after the onset of CD/UC, including diarrhea, constipation, abdominal bloating, abdominal pain, chronic pain, and abnormal weight loss in CD/UC + IBS and CD/UC cohorts. In addition, we compared the 5-year risks of developing IBD-related complications in the cohorts, including perforation of the intestine, gastrointestinal bleeding, colon cancer, gastrointestinal fistula, and abdominal abscess. The index event was defined as the onset of CD or UC in each cohort. Patients with the outcomes prior to the diagnosis of CD or UC were excluded.

### 2.4. Statistical Analysis

Chi-square analyses were performed to compare categorical parameters, and an analysis of variance (ANOVA) was performed to compare the continuous variables. Propensity score matching for baseline characteristics was performed using the TriNetX built-in algorithm, which was based on 1:1 nearest neighbor matching with a caliper of 0.1 SD. The number of patients with the outcomes and 5-year risks of developing the outcomes was obtained. Risk differences were calculated. Odds ratios (OR) with a 95% confidence interval were calculated to compare the risks of developing the outcomes between the cohorts. *p* < 0.05 was deemed to be statistically significant.

## 3. Results

### 3.1. Population

We identified a total of 5510 patients with concomitant CD and IBS (63.0% female, the average age when diagnosed with CD: 38.0 ± 8.2 years old) and 60,180 patients with CD alone (49.3% female, the average age when diagnosed with CD: 40.3 ± 19.4 years old) (Table 1A). After propensity score matching was performed, there was a total of 5389 patients in both CD + IBS and CD cohorts. There was a total of 4200 patients with concomitant UC and IBS (60.4% female, the average age when diagnosed with UC: 41.1 ± 19.5 years old) and 67,639 patients with UC alone (47.3% female, the average age when diagnosed with UC: 47.5 ± 19.3 years old) (Table 1B). After propensity score matching was performed, there was a total of 4115 patients in both UC + IBS and UC cohorts.

### 3.2. Gastrointestinal Symptoms

Patients with CD + IBS were more likely to develop gastrointestinal symptoms than patients without IBS after propensity score matching, including diarrhea (44.8 vs. 21.2%; OR 3.02, *p* < 0.0001), constipation (24.1 vs. 10.6%; OR 2.68, *p* < 0.0001), abdominal bloating (22.9 vs. 8.5%; OR 3.20, *p* < 0.0001), abdominal pain (53.7 vs. 29.1%; OR 2.83, *p* < 0.0001), chronic pain (20.9 vs. 9.1%; OR 2.65, *p* < 0.0001), and abnormal weight loss (16.5 vs. 7.1%; OR 2.57, *p* < 0.0001) (Table 2A, Figure 2A).

Similarly, patients with UC + IBS were also more likely to develop various gastrointestinal symptoms than those without IBS after propensity score matching, including diarrhea (46.8 vs. 18.9%; OR 3.77, *p* < 0.0001), constipation (21.3 vs. 8.6%; OR 2.86, *p* < 0.0001), abdominal bloating (20.4 vs. 7.1%; OR 3.32, *p* < 0.0001), abdominal pain (49.5 vs. 22.9%; OR 3.31, *p* < 0.0001), chronic pain (19.6 vs. 8.1%; OR 2.77, *p* < 0.0001), and abnormal weight loss (15.7 vs. 6.2%; OR 2.84, *p* < 0.0001) (Table 2B Figure 2B).

### 3.3. IBD-Related Complications

Patients with CD + IBS had higher risks of developing CD-related complications than patients without IBS, including perforation of the intestine (1.5 vs. 0.9%; OR 1.71, *p* = 0.0032), gastrointestinal bleeding (5.5 vs. 2.6%; OR 2.18, *p* < 0.0001), colon cancer (2.0 vs. 1.2%; OR 1.65, *p* = 0.0014), gastrointestinal fistula (3.4 vs. 2.4%; OR 1.44, *p* = 0.0019), abdominal abscess (3.0 vs. 1.7%; OR 1.75, *p* < 0.0001) (Table 2A, Figure 3A).

Patients with UC + IBS were noted to have a higher risk of developing UC-related complications than patients without IBS, including perforation of the intestine (1.3 vs. 0.3%; OR 3,77, *p* < 0.0001), gastrointestinal bleeding (7.3 vs. 2.8%; OR 2.77, *p* < 0.0001), colon cancer (1.7 vs. 0.9%; OR 1.86, *p* = 0.0022), abdominal abscess (2.0 vs. 0.8%; OR 2.52, *p* < 0.0001) (Table 2B, Figure 3B).

### 3.4. Surgical Interventions

Patients with CD + IBS had higher risks of undergoing surgical interventions than patients without IBS, including small bowel resection (1.5 vs. 0.6%; OR 2.61, *p* < 0.0001), colectomy (3.6 vs. 2.6%; OR 1.41, *p* = 0.0026), appendectomy (1.0 vs. 0.5%; OR 2.06, *p* = 0.0026), cholecystectomy (1.8 vs. 1.0%; OR 1.83, *p* = 0.0004), exploratory laparotomy (0.8 vs. 0.4%; OR 1.89, *p* = 0.011), and hysterectomy (1.0 vs. 0.4%; OR 2.27, *p* = 0.0006) (Table 2A, Figure 4A).

Similarly, patients with UC + IBS were more likely to undergo surgical interventions than those without IBS, including colectomy (3.9 vs. 2.2%; OR 1.84, *p* < 0.0001), appendectomy (1.1 vs. 0.6%; OR 1.80, *p* = 0.0195), cholecystectomy (1.6 vs. 0.8%; OR 1.98, *p* = 0.0011), exploratory laparotomy (0.8 vs. 0.2%; OR 3.22, *p* = 0.0007), and hysterectomy (0.8 vs. 0.4%; OR 2.14, *p* = 0.0103) (Table 2B, Figure 4B).

## 4. Discussion

In this large population-based study, we found that IBS is an independent risk factor for developing various gastrointestinal symptoms, IBD-related complications, and undergoing surgical interventions in patients with IBD. This is the first study to describe the risks of undergoing surgical interventions in patients with concomitant IBD and IBS, highlighting the importance of accurate diagnosis in this population.

Our study revealed that IBS is a risk factor for developing various gastrointestinal symptoms in patients with IBD, including abdominal pain, diarrhea, constipation, bloating, and abnormal weight loss. They are also at an increased risk for developing chronic pain after the onset of IBD. There are no previous studies to survey and describe the differences in symptom presentation in patients with concomitant IBD + IBS versus those with IBD alone. Accurate diagnosis is essential in these cases to ensure appropriate medical management and avoid unnecessary interventions.

Surprisingly, IBS increases the risks of developing IBD-related complications, including perforation of the intestine, gastrointestinal bleeding, colon cancer, and abdominal abscess for patients with IBD. For patients with CD, IBS also increases the risks of developing gastrointestinal fistula. The increasing risk of IBD-related complications appears to be more prominent for patients with UC. UC patients with IBS are 3.8 and 2.8 times more likely to develop gastrointestinal perforation and bleeding, respectively. Even though traditionally, IBS is considered a “functional” disease of the gastrointestinal system, IBS shares some similar mechanisms with IBD, such as disturbance in gut microbiota [15,27,28,29] and impaired gut barrier function [30], which can trigger or worsen IBD in return. In addition, disturbance of the gut-brain axis in IBS can also contribute to a worsening immune response in IBD [31,32] and result in more IBD-related complications. Overlapping IBD and IBS could represent a sicker subgroup of patients with more prominent gastrointestinal symptoms and complications. Further research is needed to determine the role of IBS in triggering or contributing to IBD-related complications and how improved diagnostic methods can help guide medical management.

IBS appears to be an independent risk factor for IBD patients to undergo surgical interventions, including colectomy, small bowel resection, cholecystectomy, appendectomy, exploratory laparotomy, and hysterectomy. One of the strong associations was observed with CD and small intestine resection, in which patients with CD + IBS are 2.6 times more likely to undergo small intestine resection. Another strong association was between UC and exploratory laparotomy, in which patients with UC + IBS were 3.2 times more likely to undergo exploratory laparotomy. This observation could potentially be explained by the fact that patients with overlapping IBD and IBS are generally more unwell and tend to have more gastrointestinal symptoms and complications. Unfortunately, there is a lack of prospective or retrospective studies in the literature regarding the risks of undergoing surgical interventions in patients with overlapping IBD and IBS. Accurate diagnosis in these patients is crucial to guide appropriate medical and surgical treatment.

Previous studies have shown that patients with IBS are at increased risk of receiving “unnecessary” surgeries [33]. Patients with IBS are 2–3 times more likely to have cholecystectomy [26,34]. It was shown that the increased risk of cholecystectomy was not due to an increased risk of gallstones but due to abdominal pain [35]. Patients with IBS are 1.6–1.7 times more likely to have a hysterectomy [26,36]. Patients with IBS are two times more likely to have appendectomies than the general population, and most patients have normal appendiceal pathology [26,37]. We observed a similar phenomenon in our study. IBD patients with overlapping IBS symptoms are at a higher risk of undergoing appendectomy, cholecystectomy, and hysterectomy. IBS symptoms can mimic organic gastrointestinal disorders. Without pathology reports, it is difficult to differentiate whether the high incidence of surgeries was due to a “true” higher prevalence of organic diseases or due to misdiagnoses since the clinical picture could be obscured by IBS symptoms. Therefore, it is essential to recognize that IBS symptoms could overlap with IBD in clinical remission, and careful clinical assessment and accurate diagnosis is critical when considering surgical interventions. 

There are several limitations to our study. In this study, the diagnoses were based on billing codes, and we cannot access an individual patient’s chart to verify the diagnoses. We cannot confirm whether the diagnoses are accurate or not since all data are de-identified and aggregated. Even though we tried to ensure that the IBS diagnosis was made during clinical remission of IBD, ESR and CRP could be nonspecific. A normal level can still be associated with active IBD. Furthermore, without access to pathology reports, it is difficult to justify the indication of the surgeries. Improved diagnostic methods could help address these limitations and better understand the relationship between IBD and IBS.

Overall, this is one of the first studies to describe the risks of developing gastrointestinal symptoms and IBD-related complications, as well as undergoing surgical interventions in patients with concomitant IBD and IBS. IBS appears to be an independent risk factor for developing various gastrointestinal symptoms, IBD-related complications, and undergoing surgical interventions in patients with IBD. Patients with concomitant IBD and IBS could represent a unique subgroup of IBD patients with more severe symptoms. Further research is warranted to determine the interaction between IBD and IBS and the importance of accurate diagnosis in guiding appropriate treatment strategies.

## Figures and Tables

**Figure 1 diagnostics-13-01901-f001:**
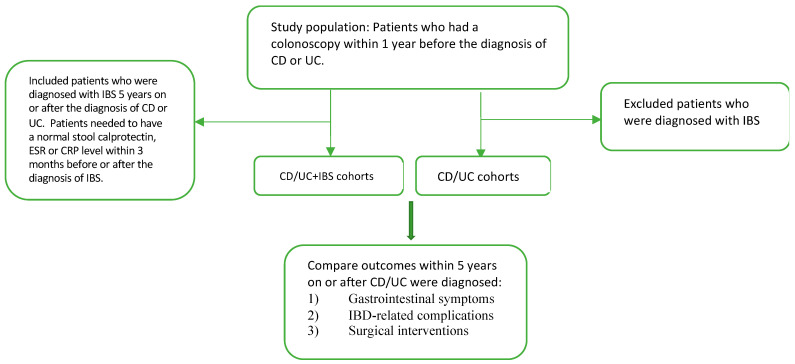
Study flow chart.

**Figure 2 diagnostics-13-01901-f002:**
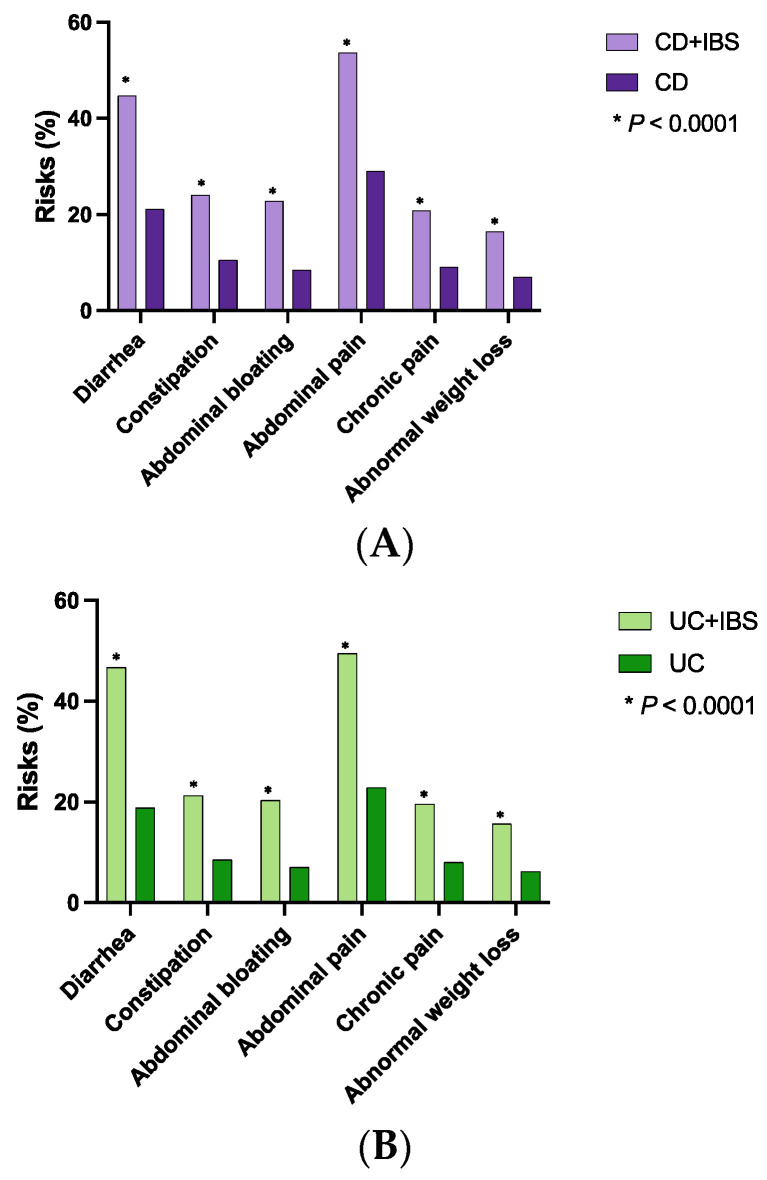
Risks of developing gastrointestinal symptoms in concomitant IBD and IBS ((**A**): Crohn’s disease and IBS, (**B**): ulcerative colitis and IBS).

**Figure 3 diagnostics-13-01901-f003:**
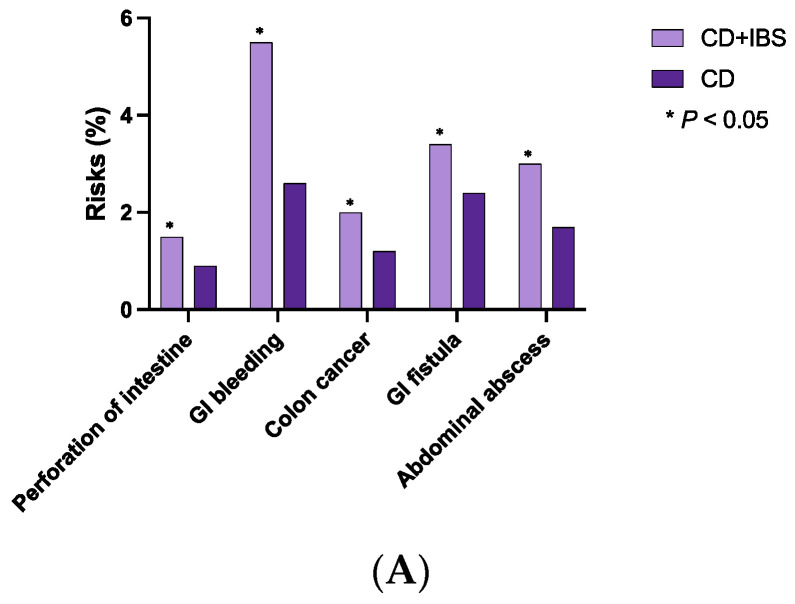
Risks of developing IBD-related complications in concomitant IBD and IBS ((**A**): Crohn’s disease and IBS, (**B**): ulcerative colitis and IBS).

**Figure 4 diagnostics-13-01901-f004:**
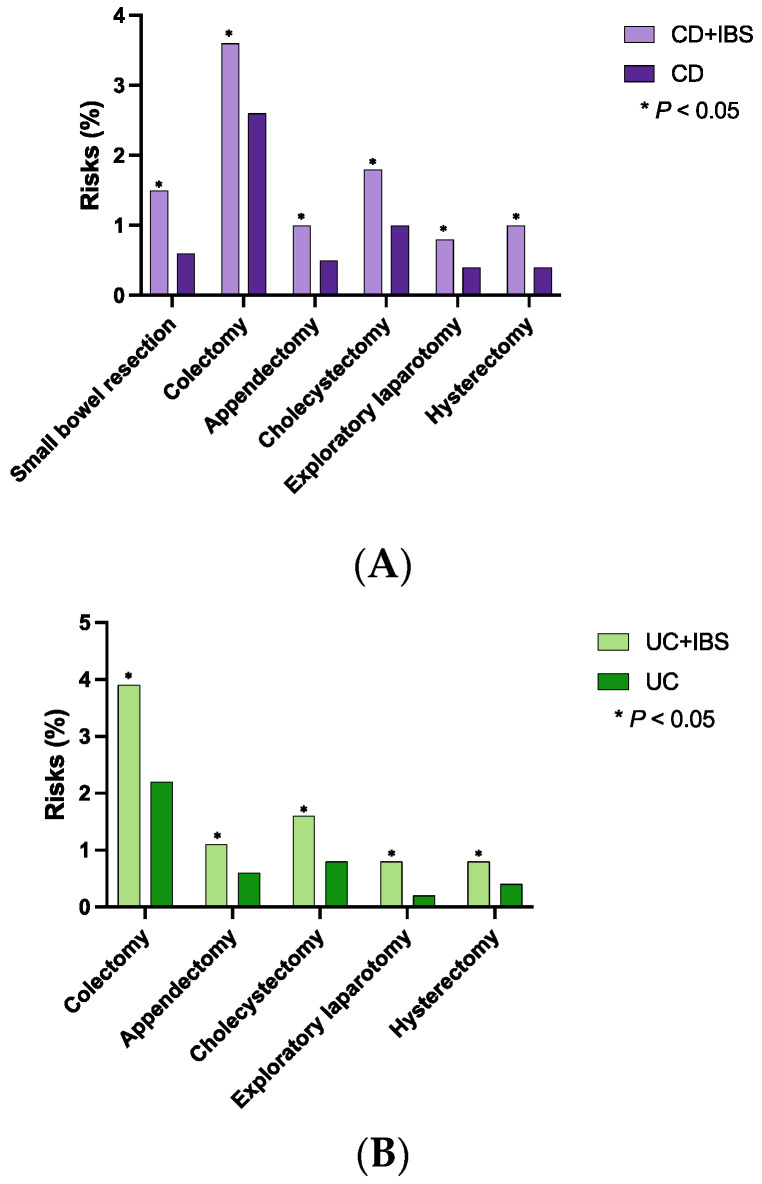
Risks of undergoing surgical interventions in concomitant IBD and IBS ((**A**): Crohn’s disease and IBS, (**B**): ulcerative colitis and IBS).

**Table 1 diagnostics-13-01901-t001:** Baseline characteristics of patients with concomitant IBD and IBS (A: Crohn’s disease and IBS, B: ulcerative colitis and IBS).

**A**									
		**Unadjusted Data**				**After Propensity-Score Matching**			
		**CD + IBS**	**CD without IBS**	***p*-Value**	**SD**	**CD + IBS**	**CD without IBS**	***p*-Value**	**SD**
N of patients		5510	60,180			5389	5389		
Average age		38.0 ± 18.2	40.3 ± 19.4	<0.001	0.122	38.0 ± 18.2	38.0 ± 18.1	0.962	0.001
Gender	Female	3396 (63.0%)	28,870 (49.3%)	<0.001	0.28	3396 (63.0%)	3392 (62.9%)	0.936	0.002
	Male	1838 (34.1%)	27,651 (47.2%)	<0.001	0.269	1838 (34.1%)	1838 (34.1%)	1.000	<0.001
Race	White	4405 (81.7%)	44,580 (76.1%)	<0.001	0.139	4405 (81.7%)	4413 (81.9%)	0.842	0.004
	Unknown	420 (7.8%)	6788 (11.6%)	<0.001	0.128	420 (7.8%)	422 (7.8%)	0.943	0.001
	Black or African American	486 (9.0%)	6144 (10.5%)	0.001	0.049	486 (9.0%)	485 (9.0%)	0.973	0.001
	Asian	54 (1.0%)	915 (1.6%)	0.001	0.05	54 (1.0%)	55 (1.0%)	0.923	0.002
Ethnicity	Hispanic or Latino	231 (4.3%)	2451 (4.2%)	0.716	0.005	231 (4.3%)	231 (4.3%)	1.000	<0.001
	Not Hispanic or Latino	4619 (85.7%)	46,235 (78.9%)	<0.001	0.179	4619 (85.7%)	4622 (85.8%)	0.934	0.002
	Unknown	539 (10.0%)	9914 (16.9%)	<0.001	0.204	539 (10.0%)	536 (9.9%)	0.923	0.002
**B**									
		**Unadjusted Data**				**After Propensity-Score Matching**			
		**UC + IBS**	**UC without IBS**	***p*-Value**	**SD**	**UC + IBS**	**UC without IBS**	***p*-Value**	**SD**
N of patients		4200	67,639			4115	4115		
Average age		41.1 ± 19.5	47.5 ± 19.3	<0.001	0.332	41.1 ± 19.5	41.2 ± 19.5	0.816	0.005
Gender	Female	2484 (60.4%)	31,284 (47.3%)	<0.001	0.264	2484 (60.4%)	2482 (60.3%)	0.964	0.001
	Male	1484 (36.1%)	31,784 (48.1%)	<0.001	0.245	1484 (36.1%)	1485 (36.1%)	0.982	0.001
Race	White	3314 (80.5%)	49,608 (75.0%)	<0.001	0.133	3314 (80.5%)	3324 (80.8%)	0.780	0.006
	Unknown	360 (8.7%)	7903 (12.0%)	<0.001	0.105	360 (8.7%)	367 (8.9%)	0.786	0.006
	Black or African American	363 (8.8%)	7010 (10.6%)	<0.001	0.06	363 (8.8%)	357 (8.7%)	0.815	0.005
	Asian	61 (1.5%)	1365 (2.1%)	0.010	0.044	61 (1.5%)	55 (1.3%)	0.575	0.012
Ethnicity	Hispanic or Latino	210 (5.1%)	3864 (5.8%)	0.049	0.033	210 (5.1%)	208 (5.1%)	0.920	0.002
	Not Hispanic or Latino	3457 (84.0%)	49,528 (74.9%)	<0.001	0.227	3457 (84.0%)	3464 (84.2%)	0.833	0.005
	Unknown	448 (10.9%)	12,737 (19.3%)	<0.001	0.236	448 (10.9%)	443 (10.8%)	0.859	0.004

**Table 2 diagnostics-13-01901-t002:** Symptoms, IBD-related complications, and surgical interventions in patients with concomitant IBD and IBS (A: Crohn’s disease and IBS, B: ulcerative colitis and IBS).

**A**								
		**CD + IBS**		**CD without IBS**		**CD + IBS vs. CD without IBS**		
		**N**	**Risk**	**N**	**Risk**	**Risk Difference**	**OR**	** *p* **
Symptoms	Diarrhea	958	44.8%	695	21.2%	23.58% (21.05–26.11%)	3.02 (2.68–3.40)	<0.0001
	Constipation	924	24.1%	482	10.6%	13.54% (11.91–15.16%)	2.68 (2.38–3.03)	<0.0001
	Abdominal bloating	976	22.9%	413	8.5%	14.42% (12.93–15.90%)	3.20 (2.83–3.62)	<0.0001
	Abdominal pain	789	53.7%	717	29.1%	24.59% (21.47–27.70%)	2.83 (2.47–3.23)	<0.0001
	Chronic pain	928	20.9%	442	9.1%	11.82% (10.38–13.27%)	2.65 (2.34–2.99)	<0.0001
	Abnormal weight loss	693	16.5%	329	7.1%	9.36% (8.01–10.70%)	2.57 (2.24–2.95)	<0.0001
Complications	Perforation of intestine	80	1.5%	47	0.9%	0.62% (0.21–1.03%)	1.71 (1.19–2.46)	0.0032
	GI bleeding	271	5.5%	133	2.6%	2.91% (2.13–3.68%)	2.18 (1.77–2.70)	<0.0001
	Colon cancer	106	2.0%	65	1.2%	0.79% (0.30–1.27%)	1.65 (1.21–2.26)	0.0014
	GI fistula	178	3.4%	125	2.4%	1.02% (0.38–1.66%)	1.44 (1.14–1.82)	0.0019
	Abdominal abscess	156	3.0%	91	1.7%	1.26% (0.68–1.84%)	1.75 (1.35–2.28)	<0.0001
Surgeries	Small bowel resection	80	1.5%	31	0.6%	0.92% (0.54–1.30%)	2.61 (1.72–3.96)	<0.0001
	Colectomy	190	3.6%	137	2.6%	1.01% (0.35–1.67%)	1.41 (1.13–1.76)	0.0026
	Appendectomy	51	1.0%	25	0.5%	0.49% (0.17–0.81%)	2.06 (1.27–3.32)	0.0026
	Cholecystectomy	93	1.8%	52	1.0%	0.80% (0.35–1.24%)	1.83 (1.30–2.58)	0.0004
	Exploratory laparotomy	45	0.8%	24	0.4%	0.39% (0.09–0.70%)	1.89 (1.15–3.10)	0.011
	Hysterectomy	54	1.0%	24	0.4%	0.57% (0.24–0.89%)	2.27 (1.40–3.68)	0.0006
**B**								
		**UC + IBS**		**UC without IBS**		**UC + IBS vs. UC without IBS**		
		**N**	**Risk**	**N**	**Risk**	**Risk Difference**	**OR**	** *p* **
Symptoms	Diarrhea	743	46.80%	469	18.90%	27.86% (24.96–30.76%)	3.77 (3.27–4.34)	<0.0001
	Constipation	627	21.30%	299	8.60%	12.69% (10.93–14.44%)	2.86 (2.47–3.32)	<0.0001
	Abdominal bloating	672	20.40%	269	7.10%	13.22% (11.62–14.82%)	3.32 (2.86–3.86)	<0.0001
	Abdominal pain	656	49.50%	497	22.90%	26.68% (23.46–29.90%)	3.31 (2.86–3.84)	<0.0001
	Chronic pain	667	19.60%	300	8.10%	11.48% (9.89–13.07%)	2.77 (2.39–3.20)	<0.0001
	Abnormal weight loss	515	15.70%	223	6.20%	9.56% (8.09–11.03%)	2.84 (2.41–3.35)	<0.0001
Complications	Perforation of intestine	52	1.30%	14	0.30%	0.94% (0.55–1.32%)	3.77 (2.09–6.81)	<0.0001
	GI bleeding	265	7.30%	105	2.80%	4.54% (3.54–5.53%)	2.77 (2.20–3.49)	<0.0001
	Colon cancer	68	1.70%	37	0.90%	0.78% (0.28–1.27%)	1.86 (1.24–2.78)	0.0022
	Abdominal abscess	79	2.00%	32	0.80%	1.17% (0.66–1.67%)	2.52 (1.66–3.80)	<0.0001
Surgeries	Colectomy	158	3.90%	88	2.20%	1.74% (0.99–2.48%)	1.84 (1.41–2.39)	<0.0001
	Appendectomy	43	1.10%	24	0.60%	0.47% (0.08–0.86%)	1.80 (1.09–2.98)	0.0195
	Cholecystectomy	66	1.60%	34	0.80%	0.81% (0.32–1.29%)	1.98 (1.31–3.00)	0.0011
	Exploratory laparotomy	32	0.80%	10	0.20%	0.54% (0.23–0.85%)	3.22 (1.58–6.56)	0.0007
	Hysterectomy	34	0.80%	16	0.40%	0.44% (0.11–0.78%)	2.14 (1.18–3.89)	0.0103

## Data Availability

Original data will be available by contacting the corresponding author.

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
