# Peer review of "Diagnostic Implications of Irritable Bowel Syndrome Is an Independent Risk Factor for Undergoing Surgical Interventions in Patients with Inflammatory Bowel Disease"

_diagnostics, 2023, doi:10.3390/diagnostics13111901_

Round 1

Reviewer 1 Report

The topic is interesting. The goal of the work is well set. Good methodology and statistical tools. The results of the work are useful. The discussion should be nourished with more references. Of the 13 references, 7 are quite outdated. The schemes are good, as are the tables. The entire manuscript would contribute to the academic community under the conditions of using even more recent references.

Author Response

Thank you for the suggestions. We have incorporated more references in the introduction and discussion.

Reviewer 2 Report

Thank you for the opportunity to review this important manuscript. Here are my comments and suggestions.

Methods section: I understand that the patients are regrouted from the database, but IBS is a diagnosis of exclusion and also has defined criteria. How did the patients get the diagnosis? Was it confirmed by testing or just labeled IBS if the patient had abdominal symptoms not from the primary IBD disease?

I am thrilled about TriNetX - a large global health research network containing de-identified aggregated electronic healthcare record (EHR) from 120 million patients worldwide. There are 93 large healthcare organizations (HCOs) in the Global Collaborative Network, most from the US and Europe. This platform provides comprehensive real-world healthcare information and allows for customized cohort selection. It contains diagnoses, procedures, medications, and laboratory values from both inpatient and outpatient settings. Interestingly large databases, even on a national level, are not so detailed. The authors could extract specific gastrointestinal symptoms from the database. This means that symptoms are put as a ''electronic field'' in that database, and I fit can be so simply extracted. Symptoms are mostly in the medical history, so one should read all these medical records, which is extremely time-consuming.

IBS has 4 major subtypes: Mostly diarrhea and abdominal discomfort (IBS-D); Mostly constipation and abdominal discomfort (IBS-C); Alternating loose stools and constipation with abdominal discomfort (IBS-mixed), and Undefined subtype (IBS-U) — symptoms vary. If this database is so detailed, then the IBS subtype could be correlated with CD and UC complications. For example, maybe only the diarrhea-predominant type results in a higher incidence of GI fistulas in CD.

I understand that there is a higher risk of cholecystectomy in IBS patients. But can the authors explain the higher risk of appendectomy because most surgical patients with UC had total (procto)colectomy, so the appendix is also removed? In CD patients, the most common location is ileocecal CD, and ileocecetomy or right hemicolectomy also includes removal of the appendix.

Author Response

1. Methods section: I understand that the patients are regrouted from the database, but IBS is a 
diagnosis of exclusion and also has defined criteria. How did the patients get the diagnosis? Was 
it confirmed by testing or just labeled IBS if the patient had abdominal symptoms not from the 
primary IBD disease?
In this database-based study, IBS diagnosis was made based on the billing codes submitted to 
the insurance company. Unfortunately, we couldn’t confirm the diagnoses ourselves since all data 
were aggregated and de-identified in this database. To ensure diagnostic accuracy, we 
specifically selected patients diagnosed with IBS when IBD is in clinical remission with negative 
inflammatory markers.
2. I am thrilled about TriNetX - a large global health research network containing de-identified 
aggregated electronic healthcare record (EHR) from 120 million patients worldwide. There are 93 
large healthcare organizations (HCOs) in the Global Collaborative Network, most from the US 
and Europe. This platform provides comprehensive real-world healthcare information and allows 
for customized cohort selection. It contains diagnoses, procedures, medications, and laboratory 
values from both inpatient and outpatient settings. Interestingly large databases, even on a 
national level, are not so detailed. The authors could extract specific gastrointestinal symptoms 
from the database. This means that symptoms are put as a ''electronic field'' in that database, and 
I fit can be so simply extracted. Symptoms are mostly in the medical history, so one should read 
all these medical records, which is extremely time-consuming.
Symptoms are commonly coded into reviews of symptoms (ROS) in the medical records and 
submitted for coding, which allowed us to extract these information.
3. IBS has 4 major subtypes: Mostly diarrhea and abdominal discomfort (IBS-D); Mostly constipation 
and abdominal discomfort (IBS-C); Alternating loose stools and constipation with abdominal 
discomfort (IBS-mixed), and Undefined subtype (IBS-U) — symptoms vary. If this database is so 
detailed, then the IBS subtype could be correlated with CD and UC complications. For example, 
maybe only the diarrhea-predominant type results in a higher incidence of GI fistulas in CD.
This database still has significant limitations and the diagnoses it contained are solely based on 
diagnostic coding: ICD codes. Unfortunately, we couldn’t really differentiate the different subtypes 
of IBS using the available database.
4. I understand that there is a higher risk of cholecystectomy in IBS patients. But can the authors 
explain the higher risk of appendectomy because most surgical patients with UC had total 
(procto)colectomy, so the appendix is also removed? In CD patients, the most common location 
is ileocecal CD, and ileocecetomy or right hemicolectomy also includes removal of the appendix.
In this study, appendectomy and colectomy are believed to be separate procedures and mutually 
exclusive. Patients underwent appendectomy should have intact colon at the time of the 
procedure. If a patient underwent total colectomy or right hemicolectomy or ileocelectomy, 
appendectomy wouldn’t be listed as a separate surgery in the medical records at the same time 
due to billing/coding reasons. 
